artificial intelligence/statistics

neural network, infectious diseases, early warning, China

**Authors for correspondence:**
Kevin He
e-mail: kevinhe@umich.edu
Dan Xiao
e-mail: danxiaoepi@163.com

# Early warning of some notifiable infectious diseases in China by the artificial neural network

Zuiyuan Guo[1], Kevin He[2] and Dan Xiao[3]

[1]Department of Disease Control, Center for Disease Control and Prevention in Northern Theater Command, Shenyang, People's Republic of China
[2]Biostatistics Department, University of Michigan, Ann Arbor, MI 48109, USA
[3]China National Clinical Research Center for Neurological Diseases, Beijing Tian Tan Hospital, No. 119, South 4th Ring Road West, Fengtai District, Beijing, People's Republic of China

ZG, 0000-0002-2784-2246

In order to accurately grasp the timing for the prevention and control of diseases, we established an artificial neural network model to issue early warning signals. The real-time recurrent learning (RTRL) and extended Kalman filter (EKF) methods were performed to analyse four types of respiratory infectious diseases and four types of digestive tract infectious diseases in China to comprehensively determine the epidemic intensities and whether to issue early warning signals. The numbers of new confirmed cases per month between January 2004 and December 2017 were used as the training set; the data from 2018 were used as the test set. The results of RTRL showed that the number of new confirmed cases of respiratory infectious diseases in September 2018 increased abnormally. The results of the EKF showed that the number of new confirmed cases of respiratory infectious diseases increased abnormally in January and February of 2018. The results of these two algorithms showed that the number of new confirmed cases of digestive tract infectious diseases in the test set did not have any abnormal increases. The neural network and machine learning can further enrich and develop the early warning theory.

## 1. Introduction

Public health emergencies caused by the continuous occurrence of emerging infectious diseases and unexplained diseases in the world such as severe acute respiratory syndrome (SARS), H1N1 influenza A, Ebola haemorrhagic fever and H7N9 highly pathogenic avian influenza have already become prominent public health issues in the past 30 years [1]. In addition, existing

infectious diseases such as tuberculosis, dengue fever, malaria and influenza are also jeopardizing human health [2–5]. Faced with the complex situation of current infectious disease prevention, we should not only strengthen the surveillance of the epidemic information of infectious diseases but also issue early warning signals as soon and as accurately as possible to strive for the timely identification of outbreaks and epidemics in the early stages and to take rapid response measures to minimize the harm to social and economic development.

Since the 2003 SARS outbreak, a web-based, daily infectious disease confirmed case surveillance system, which covered 37 notifiable infectious diseases, was instituted in April 2004 in China [1]. By 2009, the number of monitored infectious diseases increased to 39 types [6]. This system is capable of collecting the number of confirmed cases of various infectious diseases nationwide in real time and greatly increases the timeliness of infectious disease reports. Therefore, it establishes an important platform for surveillance of the epidemic intensity, analysing epidemiological patterns and practising the early warning model [7].

Early warning of infectious diseases involves issuing signals before or in the early stages of the infectious disease outbreak to warn that the event may occur or its extent and degree may expand. This is an important prevention measure to avoid or reduce harm to public health and social security caused by infectious diseases. Currently, early warning systems of infectious diseases have already been established in many developed countries; these systems include the Global Outbreak Alert and Response Network established by the WHO in 2000 [8], the Global Public Health Intelligence Network established by collaboration between the WHO and Health Canada in 1997 [9] and ProMED-mail established in 1994 and initiated by the International Society for Infectious Diseases [10]. These systems play important roles in the prevention and control of infectious diseases, particularly the prevention of emerging infectious diseases and bioterrorism attacks. The China Infectious Diseases Automated-alert and Response System developed by the Chinese Center for Disease Control and Prevention began to operate nationwide in April 2008 [11]. It is a more perfect and practical early warning system of infectious diseases established in China for the first time. It plays an irreplaceable role in the prevention and control of infectious diseases in China.

The purpose of analysing infectious disease surveillance data obtained from an early warning model is to eventually determine whether the epidemic displays any abnormal increases. Currently, commonly used early warning systems include simple control charts, moving average control charts, exponential weighted moving average and space scan statistics [7]. These systems have already played positive roles in early warnings for poliomyelitis, bacillary dysentery, measles and other public health emergencies. In the twenty-first century, research on artificial neural networks continues to deepen and includes extensive applications in many fields including precision medicine and public health [12–16]. In this study, we used real-time recurrent learning (RTRL) and extended Kalman filter (EKF) to perform early warning research on four types of respiratory infectious diseases (measles, influenza, rubella and mumps) and four types of digestive tract infectious diseases (hepatitis A, hepatitis E, typhoid fever and paratyphoid fever, and bacterial and amoebic dysentery) that have higher incidence rates among notifiable infectious diseases in China. Currently, these two algorithms have been extensively applied in nonlinear prediction and model establishment [17]. We used these technologies in the field of epidemiology. The number of new confirmed cases per month was used as a sequential input signal and nonlinearly mapped to an output signal. If an output signal was over the threshold value, it was possible to determine the abnormal increase in the number of cases in the early stage of the disease epidemic. This study enriches the existing early warning theory of infectious diseases to further excavate the application potential of public health big data and increase the level of analysis and prediction of the condition of infectious diseases.

# 2. Methods

## 2.1. Data sources

The numbers of new confirmed cases of eight types of infectious diseases reported per month between January 2004 and December 2015 were obtained from the Center for National Public Health Scientific Data (http://www.phsciencedata.cn). This center only collected data until 2015 [18]. Data between January 2016 and December 2018 were obtained by consulting the National Health Commission website (http://www.nhc.gov.cn/) [6].

## 2.2. Real-time recurrent learning

Infectious diseases have the attribute of continuous propagation in the time dimension; in other words, the numbers of new confirmed cases of one infectious disease in consecutive months are correlated. Based on this objective fact, we selected the recurrent neural network model for analysis. The so-called recurrent network indicates that the network uses the output produced after one input as an intermediate variable, which is used as a part of an input layer together with the next input vector for continuous calculation. The recurrent neural network continuously receives input signals; therefore, it is called the dynamically driving recurrent neural network. Targeting this model, we used RTRL and EKF to estimate the synaptic weights of the network [17].

The RTRL learning algorithm indicates that the adjustment of synaptic weights of a fully connected network is real-time. Figure 1$a$ shows that the recurrent network was composed of $q$ neurons and $m$ external inputs. The network had two different layers: the concatenated input-feedback layer and the processing layer for calculating nodes. Accordingly, the synaptic connections of the network were also composed of feed-forward and feedback connections. Because four types of respiratory and four types of digestive tract infectious diseases were included, the value of $m$ was 4. The number of neurons, $q$, in the processing layer was also set to 4. The output of the network was the last neuron in the processing layer.

In addition, the numbers of new confirmed cases of the four types of infectious diseases were used as inputs in the model. Because the numbers of cases of different diseases might have correlations and seasonal fluctuations [19–24], they were standardized in advance. After standardization was completed, the average value of each input variable should be approximately zero, there should be no correlation between variables, and covariances should be approximately equal. Since the learning process in this study was supervised learning, the expected responses of the network should be determined. We calculated the cumulative probabilities of 168 training samples under a multivariate normal distribution. These probabilities were arranged from low to high, and the 90% quantile was used as the threshold value. All neurons were set to have the same activation function, and a sigmoid odd function in the form of a hyperbolic tangent equation was used as the activation function. The parameters of this function were set to appropriate values such that the expected responses of the network were 1 and −1 [17], which denoted over the threshold value and lower than the threshold value, respectively. Furthermore, when the test samples were identified by our network, we consider that an early warning signal will be generated once the output value of the network is greater than zero. Moreover, in terms of the learning rate, we selected search-then-converge scheme [17]. Respiratory and digestive tract infectious diseases were separately calculated using 168 sets of numbers of cases between 2004 and 2017 as the training sets and 12 sets of numbers of cases in 2018 as the test set.

## 2.3. Extended Kalman filter

Based on the recurrent network established in figure 1$a$, the idea of the sequential state estimation was used to divide the network state space under training into the actual state and the measurement state of the system [17]. The former could not be directly observed; instead, a set of observation values was measured indirectly to estimate the actual state of the system. The entire set of synaptic weights of neurons was used as the actual state of the system. The value that measured whether the current condition of infectious diseases was at a normal level, i.e. expected response, was used as the observation state of the system. The vector activated by recurrent nodes of the network and the input vector used as the driving force together formed the input signal. The activation function was the same as that in RTRL. Therefore, the process of mapping from the input space to the output space was also nonlinear. This task was finished using EKF. Figure 1$b$ displays the basic framework of this algorithm. $w_n$ indicates the synaptic weights of the network at the $n$th time step, which was the state vector of the system. It and dynamic noise together form the state vector of the next time step of the system. The recurrent node activation vector, input vector and state vectors in the system were mapped to the output layer under the function of the nonlinear measurement function. The output vector was influenced by multivariable white noise to form the observable state of the system. The specific model establishment process is shown in the electronic supplementary material. The above algorithms are shown in the electronic supplementary material.

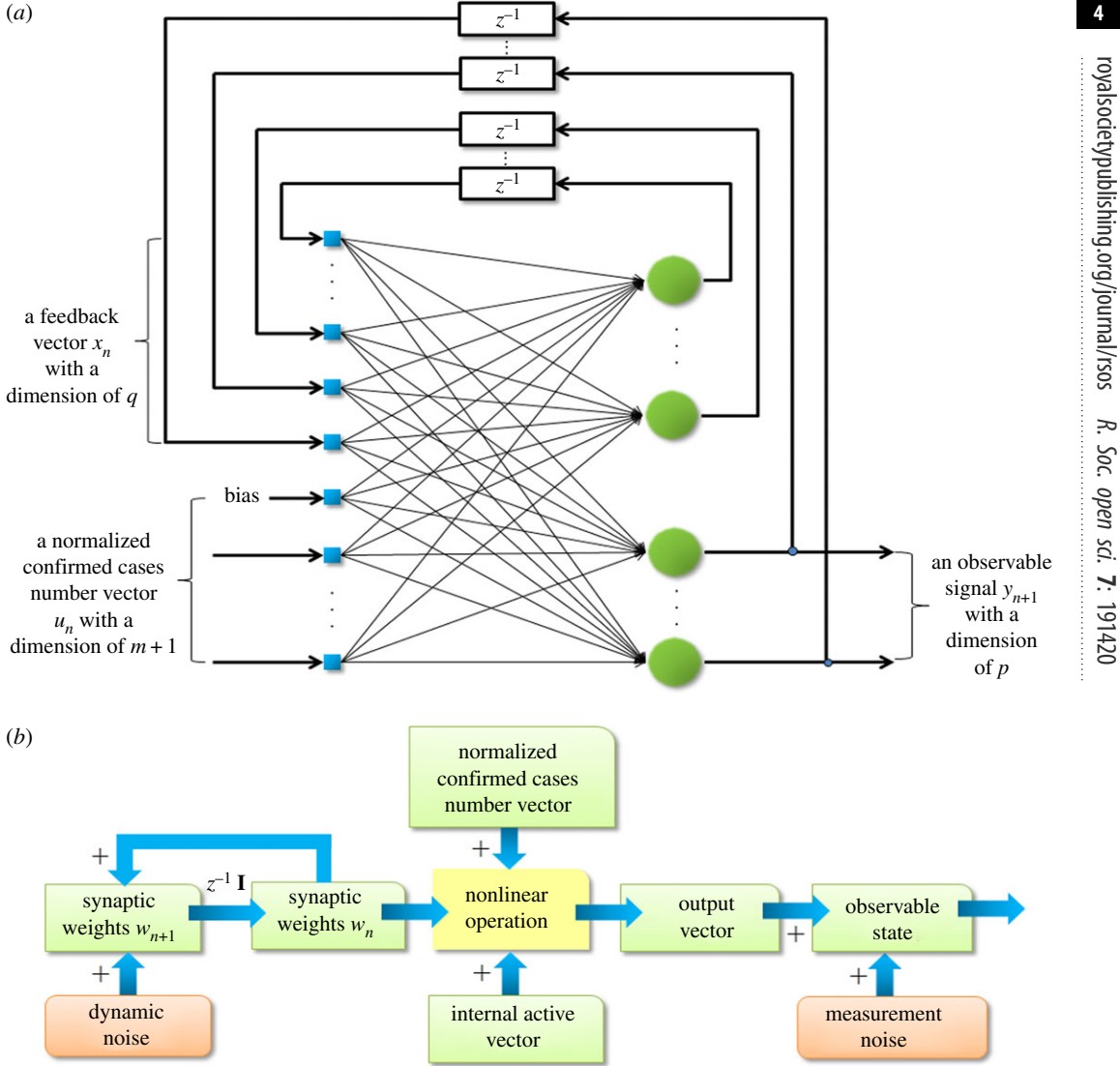

**Figure 1.** The structural framework of the real-time recurrent network and the sequential state estimation model. (*a*) The real-time recurrent neural network used for the description of real-time recurrent learning. Blue blocks indicate concatenated input-feedback layers that were composed of a state vector $x_n$ with a dimension of $q$ and an input vector $u_n$ with a dimension of $m + 1$. Green circles indicate the calculating node processing layer with a dimension of $q$; the output vector was the vector $y_{n+1}$ with a dimension of $p$. (*b*) The internal dynamic nonlinear state-space model of the recurrent network under supervised training.

## 3. Results

### 3.1. Time distribution of cases

Figure 2*a* shows the time distribution described by the surveillance data of four types of respiratory infectious diseases. The number of mumps cases increased in 2011 and 2012. The number of influenza cases increased between 2016 and 2018, particularly between December 2017 and February 2018 compared to the historical data. The numbers of measles and rubella cases showed a decreasing trend overall. Figure 2*b* shows the time distribution of the four types of digestive tract infectious diseases. They all showed obvious seasonal fluctuations. Except for hepatitis E, which had a high incidence in the spring, the other digestive tract infectious diseases all had high incidences in the summer. In addition, except for hepatitis E, the numbers of cases of the other three infectious diseases all showed a decreasing trend year by year.

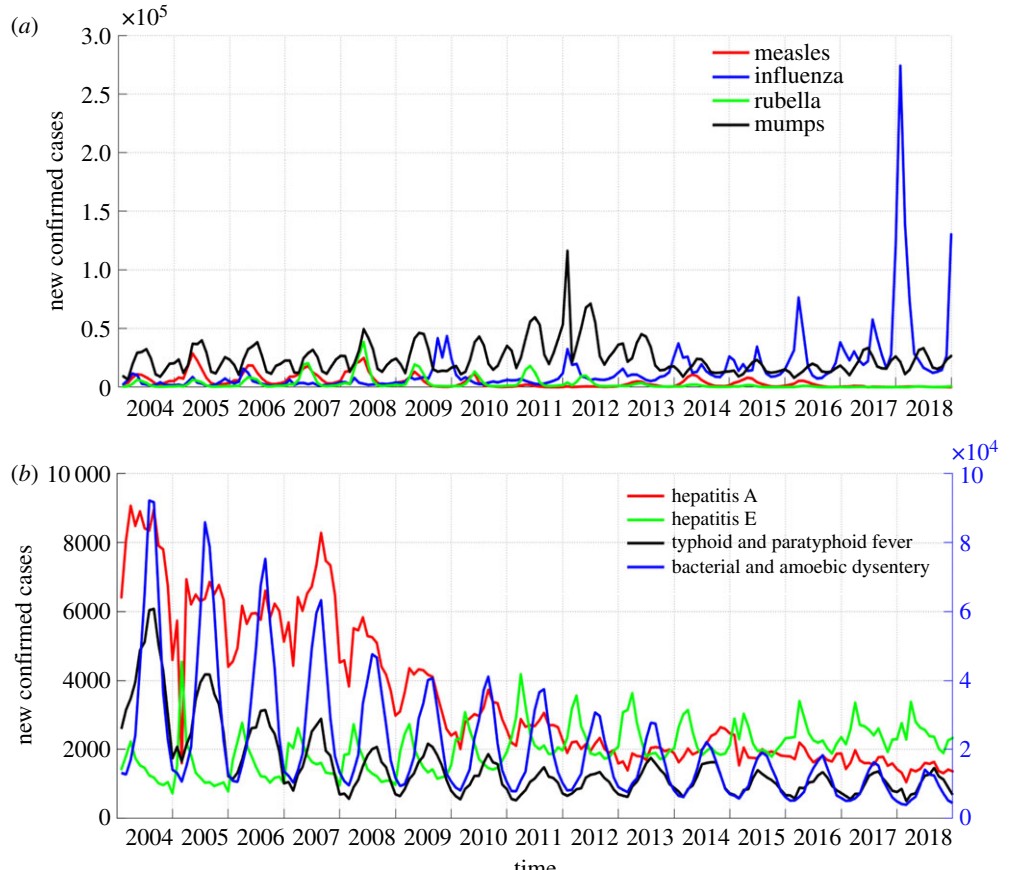

**Figure 2.** Time distribution of new confirmed cases of respiratory infectious diseases and digestive tract infectious diseases per month. (*a*) Time distribution of the number of new confirmed cases of respiratory infectious diseases per month. (*b*) Time distribution of new confirmed cases of digestive tract infectious diseases per month. The numbers of cases of the first three diseases are measured using the *y*-axis on the left, and the number of cases of the last disease is measured using the *y*-axis on the right.

## 3.2. Early warning models

The model was trained after 168 epochs, then the synaptic weights of the network were determined, and data from 2018 was used to validate model performance. Figure 3*a* shows the early warning results of respiratory infectious diseases between January and December of 2018. The RTRL results showed that the number of cases in September 2018 was significantly higher than the historical level; although the early warning value in July was higher than zero, the increase in the number of cases was not significantly different from the historical level. The EKF results showed that the numbers of cases in January and February of 2018 were significantly higher than the historical level. Figure 3*b* shows the early warning results of digestive tract infectious diseases. Both algorithms showed that the number of cases throughout all of 2018 was not significantly higher than the historical level in the same period.

Figure 4 shows the time distribution of the numbers of cases of the four types of respiratory infectious diseases in the same historical period when the early warning signal was issued. The results showed that the major reason for issuing the early warning was the increase in influenza and mumps cases. Particularly in January and February of 2018, a nationwide influenza pandemic occurred in China, which caused a significant increase in the number of influenza cases [6].

## 4. Discussion

Analyses of surveillance data using early warning models are used to eventually make a decision on whether the epidemic is increasing abnormally. Two types of calculating principles are employed for

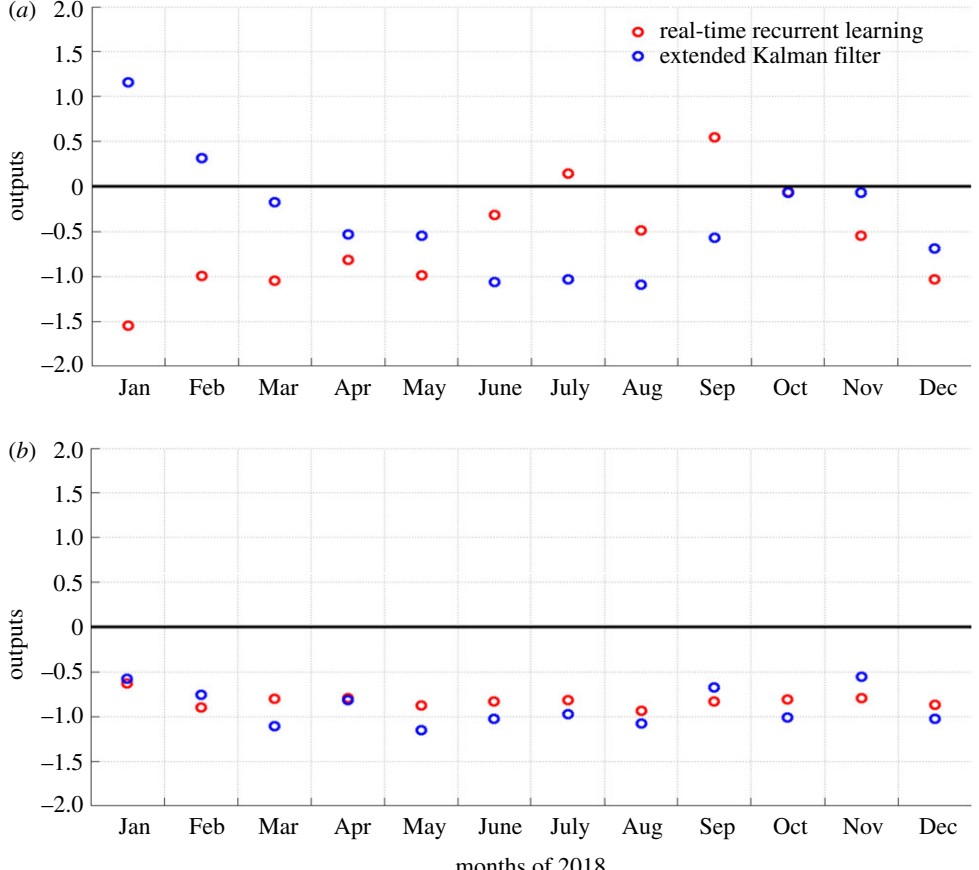

**Figure 3.** Early warning results of the model for respiratory infectious diseases and digestive tract infectious diseases between January and December of 2018. (*a*) The early warning results of respiratory infectious diseases. (*b*) The early warning results for digestive tract infectious diseases.

commonly used early warning models. One type is the calculation of the threshold value of statistics under a certain confidence level based on some specific statistical distributions and previous surveillance data; if the current statistical level obtained from actual measurement values is higher than the threshold value, the early warning signal is issued. The other type is the calculation of the expected number of cases based on specific statistical distributions and previous surveillance data; when the actual value is higher than the expected value, the early warning signal is issued [7]. For example, some classical warning approaches, such as the exponential smoothing model, Poisson regression and the ARIMA model, that follow the above ideas have been effectively applied and constantly improved to increase the accuracy of early warning [7]. The innovation of this article is that we broke through the traditional thinking framework. We did not calculate the threshold value or the expected number of cases; instead, the system itself calculated the epidemic intensity and made the decision on whether to issue early warning signals. The neural network has the characteristics of continuous optimization through self-iteration; therefore, when there are more training times on the system, its decision will be more accurate and the system will present a certain intelligent type. Furthermore, this method can be used not only to consider multiple types of infectious diseases at the same time, but also to analyse multivariate data, such as the number of patients in outpatient clinics, drug consumption data in hospitals and sales data for over-the-counter drugs in pharmacies. By analysing the complex relationship among multiple factors, we can judge the law of disease development and changes comprehensively.

Both respiratory and digestive tract infectious diseases have different epidemic patterns in different seasons. The former mainly has epidemics in the winter and spring and the latter mainly has epidemics in the summer. Therefore, the standardized processing of input vectors of the model should consider the influence of seasonal factors. According to the epidemic patterns of infectious diseases, we believed that vectors consisting of the numbers of cases of the four types of respiratory infectious diseases or the four types of digestive tract infectious diseases in every month all

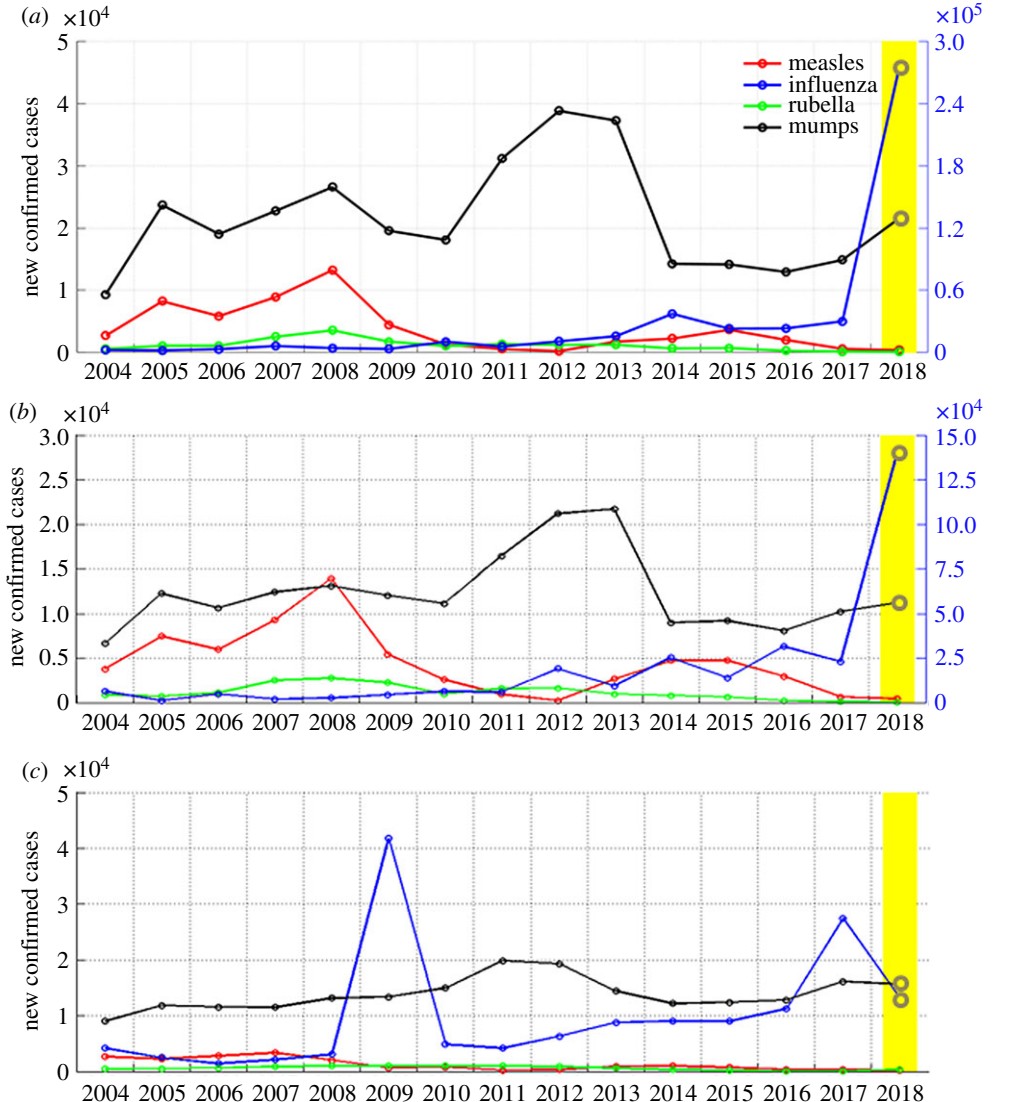

**Figure 4.** Historical data of the numbers of cases in the same period of months issuing an early warning of respiratory infectious diseases. (a–c) Historical numbers of cases of the four types of respiratory infectious diseases in January, February and September since 2004.

conformed to the multivariate normal distribution; in addition, the average vector and covariance matrix in every month also differed. After the standardized processing of all input vectors, the influence of seasonal factors on this model could be eliminated. We used month as a unit to perform analyses on the numbers of cases in this study. To improve the timeliness of early warning, week or day can be used as the unit for analysis. The average vectors and covariance matrix should also be adjusted accordingly.

The level of the threshold value could be adjusted according to the actual condition. When the threshold value was higher, the number of cases necessary to issue an early warning was higher; when the threshold value was lower, the result was the opposite. The learning rate determined the convergence rate of the network model. If the learning rate was a constant number, the convergence rate was slower. If the learning rate was a constant divided by the times of iteration, although the convergence of the stochastic approximation algorithm could be ensured, fewer iteration times might have the risk of parameter amplification when the constant value was higher [25,26]. Therefore, we chose the search-then-converge schedule as the commonly used learning rate annealing programme in online learning. It had the advantage of combining the expected characteristics of the model with the traditional stochastic approximation theory [27].

In this study, the numbers of cases of some types of infectious diseases were used as the input variables in the model to make an early warning decision through comprehensive analysis. Therefore, an early warning was issued only when the numbers of cases of these infectious diseases showed an overall increase and exceeded the threshold value. However, it was not necessary to issue an early warning only when all numbers of cases increased. Figure 4 shows that when the system issued an early warning, the numbers of measles and rubella cases not only did not increase but also were at lower levels compared to the historical level in the same period. The model made the decision on the high incidence of respiratory infectious diseases after the numbers of influenza and mumps cases increased. Therefore, we should perform the analysis on the actual number of cases of each infectious disease based on the model result to determine which infectious disease epidemic caused the early warning. If the model did not issue an early warning, then it could not indicate that the number of any infectious disease cases did not increase significantly. Thus, during the analysis of the epidemic, we should assess actual data for a detailed analysis. If the analysis was performed only based on the results of the model, it would be possible that the number of some infectious diseases already significantly increased but the model did not timely issue an early warning, thus causing delay in the timing of prevention and control.

According to figure 3, we found that the proportion of the same results according to the two algorithms is 83.3% (20/24), indicating that their decisions regarding the epidemic trend of diseases were similar. Different results occurred in January, February, July and September of 2018. According to figure 4a,b, only the numbers of influenza cases in January and February of 2018 greatly increased compared to that in the historical periods; conversely, the numbers of cases of the other three infectious diseases all remained at lower levels. Figure 4c shows that the numbers of mumps and influenza cases slightly increased in September 2018. At these three time points, the results of these two algorithms differed. These results indicated that although the numbers of cases increased, the overall results were not significantly higher than that in the same historical periods. The early warning value of RTRL in July was slightly higher than zero; thus, we ignored its early warning significance. To increase the sensitivity of early warning, these two methods could be used simultaneously for analysis and determination of the epidemic intensity of the disease through a comparison of early warning results between these two algorithms.

In this study, we selected eight types of infectious diseases with a higher incidence for analysis. These two models are not applicable for diseases with lower incidences such as anthrax, cholera and the plague; therefore, other methods are necessary for early warning [7]. Because the Chinese Notifiable Infectious Diseases Surveillance and Report System began to operate nationwide in 2004 [18], only disease data in the recent 16 years could be obtained. The sample size of the training set was small, which may have influenced the calculation results for synaptic weights in the model to a certain extent. This study only references a new modelling approach and was limited in its scope in terms of validation. Future work will be focused on gathering more data and cases and providing a more comprehensive model validation. Furthermore, we evaluated the epidemic intensities of infectious diseases in the entire nation. However, infectious disease epidemics, such as dengue fever and malaria, might be limited to a certain area, or they may spread throughout the entire country after an outbreak in a certain area, such as SARS and H1N1 influenza A. If analysis and judgement were performed only based on the national data of these diseases, prevention and control of the epidemic would usually be delayed. Therefore, this model should be implemented in every province and every city to increase the timeliness of early warning of infectious diseases all over the country. In addition, real-time multivariate analysis is currently one of the research focuses in the field of early warnings for diseases, including multistatistical process control, biological change-point detection and some others [7]. Since the application of artificial neural networks in the field of early warnings for diseases is relatively new, further research is required to achieve greater precision.

Ethics. Since there is no experiment of humans and animals in the study, we did not require review by the ethical institutional review board or written informed consent.

Data accessibility. The raw data required for this study are available in the electronic supplementary material.

Authors' contributions. Z.G. conceived of the study, designed the study, analysed the data and drafted the manuscript; K.H. provided guidance regarding the algorithm; D.X. revised the manuscript and provided comments. All authors gave final approval for publication.

Competing interests. All authors declare no competing interests.

Funding. This work was supported by the National Science and Technology Major Project (grant no. 2018ZX10713003) and the National Key R&D Program of China (grant no. 2018YFC1311703).

Acknowledgements. We would like to thank American Journal Experts (www.aje.com) for English language editing.

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
