## [Reviewer comments · Royal Society Open Science]

Review History

RSOS-191420.R0 (Original submission)

Review form: Reviewer 1

Is the manuscript scientifically sound in its present form?

No

Are the interpretations and conclusions justified by the results?

Yes

Is the language acceptable?

No

Do you have any ethical concerns with this paper?

No

Have you any concerns about statistical analyses in this paper?

No

Recommendation?

Major revision is needed (please make suggestions in comments)

Comments to the Author(s)

Paper overview

This paper summarizes the use of artificial neural networks to provide early warning in the event of a potential increase incidence or outbreak of an infectious disease. The manuscript discusses leveraging national databases which update dynamically to drive the ultimate implementation of the modeling approach as a surveillance and monitoring tool. The group uses a recurrent neural network that leverages real-time recurrent learning and an extended Kalman filter to estimate and adapt model weights dynamically. Implementation of this approach will help provide early warning in the event of an outbreak and allow local and government resources to respond in a timely manner.

General Comments:

The paper would benefit significantly by a more organized description of their approach and results. There are some instances in the paper where methods and results are extremely difficult to follow, especially with the figures and graphical results provided. The overall readability of this manuscript should be significantly improved such that the methodology and results are more easily followed and understood. Furthermore, perhaps there needs to be sufficient justification as to why this approach is best suited rather than other predictive modeling approaches. Other more specific comments are referenced below that may significantly improve the readability of the manuscript:

- 1.) Make synaptic weights plural in the last sentence of the first paragraph in the section of Real-time recurrent learning.
- 2.) Under the Methods section, the description of the model inputs should be enhanced to be better understandable by the target audience. In its current state it takes a lot of time to interpret the text to determine what the model inputs are. The figures are too generalized. Recommend restructuring Figures 1A and 1B to be more consistent with the text and adding information on external model inputs and data flow through the model as described in the text as it relates to specific model data features/vectors (e.g. the number historical and newly confirmed cases of a specific infectious disease type). Make the Figures less general and more aligned with specific approach as referenced in the text. Improve the figures to visually demonstrate the use of real-time recurrent learning and extending Kalman filtering in the weight estimation for the recurrent neural network.
- 3.) There is reference in the text to supplemental materials that demonstrate the model establishment process as well as algorithms, however, no supplemental materials are provided. These may address some of the items discussed above, however, it cannot be determined given the absence of this information.
- 4.) The Figures in the Results section should be improved significantly, they should have legends embedded within the Figure images to label the various line plots and what they correspond to. Furthermore, the results should be described in more supporting detail. From the text it is hard to follow why the authors chose the output scale on Figure 3A and 3B to be between -2 and 2, if a hyperbolic tangent transfer function is used it should normalize model inputs and outputs to be between -1 and 1. A justification for this should be provided.
- 5.) The authors should provide a better explanation on how the model determines and makes the decision (deciding threshold) to execute the early warning/alert this wasn't explained especially clearly in the text and discussion section.
- 6.) In the discussion section, alternative modeling approaches should be suggested that may be perhaps better suited to the targeted end use case of this approach. For example, using incidence

of events as a time series and forecasting future trends over N months in advance. For example, can autoregressive modeling of each incidence of event be incorporated just as effectively, or time delay neural network models? If this approach is superior additional section in the discussion section should provide significant justification as to why.

7.) The proposed modeling approach uses a rather limited set of input features to derive its prediction the authors should acknowledge this as another limitation and suggest other potential model input features that could support improved model performance in the future. The authors do acknowledge seasonal factors as a potential factor and that the model compensates from this in its various state/input vectors spanning a year. A major benefit of ANNs such as this is the ability to accommodate multiple model input factors and predictors, there is likely other model predictors that would provide more predictive accuracy and the authors should identify them as future pursuits.

8.) The last paragraph on 15 that extends to 16 needs to be rephrased to be more concise and to enhance clarity and as it is difficult to follow as it relates to non-specific nature of the alerts. This should also be referenced in the Results section and Figure 4 should be improved to highlight the alerts being attributed to the increased incidence of influenza and mumps cases by circling or marking this in the Figure.

Decision letter (RSOS-191420.R0)

07-Oct-2019

Dear Dr Guo,

The editors assigned to your paper ("Early warning of some notifiable infectious diseases in China by the artificial neural network") have now received comments from reviewers. We would like you to revise your paper in accordance with the referee and Associate Editor suggestions which can be found below (not including confidential reports to the Editor). Please note this decision does not guarantee eventual acceptance.

Please submit a copy of your revised paper before 30-Oct-2019. Please note that the revision deadline will expire at 00.00am on this date. If we do not hear from you within this time then it will be assumed that the paper has been withdrawn. In exceptional circumstances, extensions may be possible if agreed with the Editorial Office in advance. We do not allow multiple rounds of revision so we urge you to make every effort to fully address all of the comments at this stage. If deemed necessary by the Editors, your manuscript will be sent back the original reviewer for assessment -- indeed, the Editors have indicated that inviting a second reviewer is likely. If the original reviewers are not available, we may invite new reviewers.

- Data accessibility

If you wish to submit your supporting data or code to Dryad (<http://datadryad.org/>), or modify your current submission to dryad, please use the following link:
<http://datadryad.org/submit?journalID=RSOS&manu=RSOS-191420>

- Competing interests

- Authors' contributions

- Acknowledgements

- Funding statement

on behalf of Dr Francois Fages (Associate Editor) and Marta Kwiatkowska (Subject Editor)
openscience@royalsociety.org

Associate Editor's comments (Dr Francois Fages):

Associate Editor: 1

Comments to the Author:

Dear authors

Please find enclosed a peer review of your paper that points to several problems in your communication and suggests a major revision for better comparison with other work.

You are invited to take into consideration those criticisms and recommendations for submitting a revised version of your paper.

Best regards

Comments to Author:

Reviewers' Comments to Author:

Reviewer: 1

Comments to the Author(s)

Paper overview

This paper summarizes the use of artificial neural networks to provide early warning in the event of a potential increase incidence or outbreak of an infectious disease. The manuscript discusses leveraging national databases which update dynamically to drive the ultimate implementation of the modeling approach as a surveillance and monitoring tool. The group uses a recurrent neural network that leverages real-time recurrent learning and an extended Kalman filter to estimate and adapt model weights dynamically. Implementation of this approach will help provide early warning in the event of an outbreak and allow local and government resources to respond in a timely manner.

General Comments:

The paper would benefit significantly by a more organized description of their approach and results. There are some instances in the paper where methods and results are extremely difficult to follow, especially with the figures and graphical results provided. The overall readability of this manuscript should be significantly improved such that the methodology and results are more easily followed and understood. Furthermore, perhaps there needs to be sufficient justification as to why this approach is best suited rather than other predictive modeling approaches. Other more specific comments are referenced below that may significantly improve the readability of the manuscript:

1.) Make synaptic weights plural in the last sentence of the first paragraph in the section of Real-time recurrent learning.

- 2.) Under the Methods section, the description of the model inputs should be enhanced to be better understandable by the target audience. In its current state it takes a lot of time to interpret the text to determine what the model inputs are. The figures are too generalized. Recommend restructuring Figures 1A and 1B to be more consistent with the text and adding information on external model inputs and data flow through the model as described in the text as it relates to specific model data features/vectors (e.g. the number historical and newly confirmed cases of a specific infectious disease type). Make the Figures less general and more aligned with specific approach as referenced in the text. Improve the figures to visually demonstrate the use of real-time recurrent learning and extending Kalman filtering in the weight estimation for the recurrent neural network.
- 3.) There is reference in the text to supplemental materials that demonstrate the model establishment process as well as algorithms, however, no supplemental materials are provided. These may address some of the items discussed above, however, it cannot be determined given the absence of this information.
- 4.) The Figures in the Results section should be improved significantly, they should have legends embedded within the Figure images to label the various line plots and what they correspond to. Furthermore, the results should be described in more supporting detail. From the text it is hard to follow why the authors chose the output scale on Figure 3A and 3B to be between -2 and 2, if a hyperbolic tangent transfer function is used it should normalize model inputs and outputs to be between -1 and 1. A justification for this should be provided.
- 5.) The authors should provide a better explanation on how the model determines and makes the decision (deciding threshold) to execute the early warning/alert this wasn't explained especially clearly in the text and discussion section.
- 6.) In the discussion section, alternative modeling approaches should be suggested that may be perhaps better suited to the targeted end use case of this approach. For example, using incidence of events as a time series and forecasting future trends over N months in advance. For example, can autoregressive modeling of each incidence of event be incorporated just as effectively, or time delay neural network models? If this approach is superior additional section in the discussion section should provide significant justification as to why.
- 7.) The proposed modeling approach uses a rather limited set of input features to derive its prediction the authors should acknowledge this as another limitation and suggest other potential model input features that could support improved model performance in the future. The authors do acknowledge seasonal factors as a potential factor and that the model compensates from this in its various state/input vectors spanning a year. A major benefit of ANNs such as this is the ability to accommodate multiple model input factors and predictors, there is likely other model predictors that would provide more predictive accuracy and the authors should identify them as future pursuits.
- 8.) The last paragraph on 15 that extends to 16 needs to be rephrased to be more concise and to enhance clarity and as it is difficult to follow as it relates to non-specific nature of the alerts. This should also be referenced in the Results section and Figure 4 should be improved to highlight the alerts being attributed to the increased incidence of influenza and mumps cases by circling or marking this in the Figure.

Author's Response to Decision Letter for (RSOS-191420.R0)

See Appendix A.

RSOS-191420.R1 (Revision)

Review form: Reviewer 1

Is the manuscript scientifically sound in its present form?

Yes

Are the interpretations and conclusions justified by the results?

Yes

Is the language acceptable?

Yes

Do you have any ethical concerns with this paper?

No

Have you any concerns about statistical analyses in this paper?

No

Recommendation?

Accept with minor revision (please list in comments)

Comments to the Author(s)

Thank you to the authors for addressing previous comments, I believe the paper's readability has been significantly improved by additional information and description being referenced in the manuscript as well as modifications to the Figures.

There are still some minor issues with language and grammar that are present in the manuscript. I'd recommend reading through and addressing some of the issues such as the example below.

On Page 12 under Early warning models, the first sentence ends abruptly and doesn't finish. It reads "After 168 iterations of the network, we determined the synaptic weights of the network and then input the test data for 2018 into the network which yielded the following results." This sentence should either use a colon "yielded the following results:" or be broken up into multiple sentences. For example, "The model was trained after 168 epochs, and data from 2018 was used to validate model performance.". The authors can then provide their summary and description of the results as is.

On page 19 the author's state: "We should continue to train the network by adding new cases in the future to optimize early warning performance". I believe there should be an expansion of this statement and the authors should highlight that this manuscript only references a new modeling approach and was limited in its scope in terms of validation. Future work will be focused on gather more data and cases and providing a more comprehensive model validation. Results in this paper should be interpreted cautiously, however, the approach should be investigated in future work.

Decision letter (RSOS-191420.R1)

20-Dec-2019

Dear Dr Guo:

On behalf of the Editors, I am pleased to inform you that your Manuscript RSOS-191420.R1 entitled "Early warning of some notifiable infectious diseases in China by the artificial neural network" has been accepted for publication in Royal Society Open Science subject to minor revision in accordance with the referee suggestions. Please find the referees' comments at the end of this email.

The reviewers and Subject Editor have recommended publication, but also suggest some minor revisions to your manuscript. Therefore, I invite you to respond to the comments and revise your manuscript.

- Ethics statement

- Data accessibility

If you wish to submit your supporting data or code to Dryad (<http://datadryad.org/>), or modify your current submission to dryad, please use the following link:
<http://datadryad.org/submit?journalID=RSOS&manu=RSOS-191420.R1>

- Competing interests

- Authors' contributions

AB carried out the molecular lab work, participated in data analysis, carried out sequence alignments, participated in the design of the study and drafted the manuscript; CD carried out the statistical analyses; EF collected field data; GH conceived of the study, designed the study,

coordinated the study and helped draft the manuscript. All authors gave final approval for publication.

- Acknowledgements

- Funding statement

Because the schedule for publication is very tight, it is a condition of publication that you submit the revised version of your manuscript before 29-Dec-2019. Please note that the revision deadline will expire at 00.00am on this date. If you do not think you will be able to meet this date please let me know immediately.

on behalf of Dr Francois Fages (Associate Editor) and Marta Kwiatkowska (Subject Editor)
openscience@royalsociety.org

Reviewer comments to Author:

Reviewer: 1

Comments to the Author(s)

Thank you to the authors for addressing previous comments, I believe the paper's readability has been significantly improved by additional information and description being referenced in the manuscript as well as modifications to the Figures.

There are still some minor issues with language and grammar that are present in the manuscript. I'd recommend reading through and addressing some of the issues such as the example below.

On Page 12 under Early warning models, the first sentence ends abruptly and doesn't finish. It reads "After 168 iterations of the network, we determined the synaptic weights of the network and then input the test data for 2018 into the network which yielded the following results." This sentence should either use a colon "yielded the following results:" or be broken up into multiple sentences. For example, "The model was trained after 168 epochs, and data from 2018 was used to validate model performance.". The authors can then provide their summary and description of the results as is.

On page 19 the author's state: "We should continue to train the network by adding new cases in the future to optimize early warning performance". I believe there should be an expansion of this statement and the authors should highlight that this manuscript only references a new modeling approach and was limited in its scope in terms of validation. Future work will be focused on gather more data and cases and providing a more comprehensive model validation. Results in this paper should be interpreted cautiously, however, the approach should be investigated in future work.

Decision letter (RSOS-191420.R2)

21-Jan-2020

Dear Dr Guo,

It is a pleasure to accept your manuscript entitled "Early warning of some notifiable infectious diseases in China by the artificial neural network" in its current form for publication in Royal Society Open Science. The comments of the reviewer(s) who reviewed your manuscript are included at the foot of this letter.

on behalf of Dr Francois Fages (Associate Editor) and Marta Kwiatkowska (Subject Editor)
openscience@royalsociety.org

Appendix A

Dear Professor Andrew Dunn,

Thank you for the recent review of our manuscript “Early warning of some notifiable infectious diseases in China by the artificial neural network”. We appreciate the diligent efforts of the editor and reviewers to help improve our manuscript. We have revised our manuscript based on the reviewers’ comments and the format requirements of the journal and wish to resubmit it for your consideration. Changes made in response to the concerns raised by the reviewers are indicated using the track changes feature in the revised manuscript. We have invited professional language experts at American Journal Experts to review the manuscript and revise the language.

All authors reviewed the final version of the revised manuscript and have approved it for publication. The manuscript has not been and will not be sent elsewhere for possible publication as long as it is under consideration by Royal Society Open Science. We hope that the revisions are acceptable and look forward to hearing from you soon.

Sincerely,

Zuiyuan Guo

No. 6, Longshan Road, 110034, Shenyang, China

Phone number: +86-18604018769

Email: zuiyuanguo@163.com

Dear Reviewers,

Thank you very much for your letter and advice regarding our manuscript entitled “Early warning of some notifiable infectious diseases in China by the artificial neural network”. We have revised our manuscript based on the comments and wish to resubmit it for your consideration. Changes made in response to the concerns are indicated using the track changes feature in the revised manuscript. We have invited professional language experts at American Journal Experts to revise the language.

Sincerely,

Zuiyuan Guo

Email: zuiyuanguo@163.com

We would like to express our sincere gratitude to the reviewers for their constructive comments.

Replies to the Reviewers

1. Make synaptic weights plural in the last sentence of the first paragraph in the section of Real-time recurrent learning.

Answer: I apologize for this mistake, which I have corrected (manuscript with tracked changes, page 6, line 12).

2. Under the Methods section, the description of the model inputs should be enhanced to be better understandable by the target audience. In its current state it takes a lot of time to interpret the text to determine what the model inputs are. The figures are too generalized. Recommend restructuring Figures 1A and 1B to be more consistent with the text and adding information on external model inputs and data flow through the model as described in the text as it relates to specific model data features/vectors (e.g. the number historical and newly confirmed cases of a specific infectious disease type). Make the Figures less general and more aligned with specific approach as referenced in the text. Improve the figures to visually demonstrate the use of real-time recurrent learning and extending Kalman filtering in the weight estimation for the recurrent neural network.

Answer: Thank you for your valuable suggestion. We have revised Figures 1A and 1B to be more specific and consistent with the approach as referenced in the text.

3. There is reference in the text to supplemental materials that demonstrate the model establishment process as well as algorithms, however, no supplemental materials are provided. These may address some of the items discussed above, however, it cannot be determined given the absence of this information.

Answer: I apologize for not providing the supplemental material, which I have submitted to the journal.

4. The Figures in the Results section should be improved significantly, they should have legends embedded within the Figure images to label the various line plots and what they correspond to. Furthermore, the results should be described in more supporting detail. From the text it is hard to follow why the authors chose the output scale on Figure 3A and 3B to be between -2 and 2, if a hyperbolic tangent transfer function is used it should normalize model inputs and outputs to be between -1 and 1. A justification for this should be provided.

Answer: We have embedded labels within the figures in the results section to denote the various line plots and their corresponding meanings. The observed signal scales of real-time recurrent learning and extended Kalman filtering are between -2 and 2 because we modified the parameters of the hyperbolic tangent transfer function to cause the mean values of the expected outputs to be -1 and 1 according to the book *Haykin S. Neural networks and learning machines, third ed. Englewood: Prentice Hall, 2009*. The specific descriptions are provided in the supplemental material (supplemental material, page 5, lines 8-19; page 6, lines 1-5).

5. The authors should provide a better explanation on how the model determines and makes the decision (deciding threshold) to execute the early warning/alert this wasn't explained especially clearly in the text and discussion section.

Answer: We have explained how the model decides to execute an early warning in detail in the methods section (page 7, lines 6-18) and in the results section (page 12, lines 2-4).

6. In the discussion section, alternative modeling approaches should be suggested that may be perhaps better suited to the targeted end use case of this approach. For example, using incidence of events as a time series and forecasting future trends over N months in advance. For example, can autoregressive modeling of each incidence of event be incorporated just as effectively, or time delay neural network models? If this approach is superior additional section in the discussion section should provide significant justification as to why.

Answer: Some alternative modeling approaches have been referenced in the first paragraph of the discussion section (page 15, lines 13-16), and we have described

some unique advantages of our model at the end of this paragraph (page 15, lines 22-24; page 16, lines 1-3).

7. The proposed modeling approach uses a rather limited set of input features to derive its prediction the authors should acknowledge this as another limitation and suggest other potential model input features that could support improved model performance in the future. The authors do acknowledge seasonal factors as a potential factor and that the model compensates from this in its various state/input vectors spanning a year. A major benefit of ANNs such as this is the ability to accommodate multiple model input factors and predictors, there is likely other model predictors that would provide more predictive accuracy and the authors should identify them as future pursuits.

Answer: We have added more details on how to overcome the limitation of training samples (page 19, lines 6-7). Some other model predictors that may provide greater predictive accuracy have been referenced at the end of the discussion section (page 19, 15-20).

8. The last paragraph on 15 that extends to 16 needs to be rephrased to be more concise and to enhance clarity and as it is difficult to follow as it relates to non-specific nature of the alerts. This should also be referenced in the Results section and Figure 4 should be improved to highlight the alerts being attributed to the increased incidence of influenza and mumps cases by circling or marking this in the Figure.

Answer: We have moved some contents to the methods section for clarification (page 16, lines 18-23; page 17, lines 2-4). Further, we have added a description of the iterations to the methods section (page 12, lines 2-4). In addition, Figure 4 has been improved to highlight the alerts with grey circles (page 14, Figure 4).